# The Use of Neural Networks and Genetic Algorithms to Control Low Rigidity Shafts Machining

**DOI:** 10.3390/s20174683

**Published:** 2020-08-19

**Authors:** Antoni Świć, Dariusz Wołos, Arkadiusz Gola, Grzegorz Kłosowski

**Affiliations:** 1Faculty of Mechanical Engineering, Lublin University of Technology, 20-618 Lublin, Poland; a.swic@pollub.pl (A.Ś.); d.wolos@pollub.pl (D.W.); a.gola@pollub.pl (A.G.); 2Faculty of Management, Lublin University of Technology, 20-618 Lublin, Poland

**Keywords:** process control, machine learning, neural networks, genetic algorithms, turning, low rigidity

## Abstract

The article presents an original machine-learning-based automated approach for controlling the process of machining of low-rigidity shafts using artificial intelligence methods. Three models of hybrid controllers based on different types of neural networks and genetic algorithms were developed. In this study, an objective function optimized by a genetic algorithm was replaced with a neural network trained on real-life data. The task of the genetic algorithm is to select the optimal values of the input parameters of a neural network to ensure minimum deviation. Both input vector values and the neural network’s output values are real numbers, which means the problem under consideration is regressive. The performance of three types of neural networks was analyzed: a classic multilayer perceptron network, a nonlinear autoregressive network with exogenous input (NARX) prediction network, and a deep recurrent long short-term memory (LSTM) network. Algorithmic machine learning methods were used to achieve a high level of automation of the control process. By training the network on data from real measurements, we were able to control the reliability of the turning process, taking into account many factors that are usually overlooked during mathematical modelling. Positive results of the experiments confirm the effectiveness of the proposed method for controlling low-rigidity shaft turning.

## 1. Introduction

About half of all parts used in different types of machinery and mechanical devices are rotating parts. They include gears, cylinders, bushings, discs, and hubs. Most of those rotating parts (approximately 40%) are various types of shafts, of which about 12% are low-rigidity shafts. There is no single accepted definition of low-rigidity shafts; however, this category is commonly taken to include shafts with a length-to-diameter ratio of no less than 10. This means that these parts have irregular, strongly elongated shapes. Low-rigidity shafts are used in the electromechanical, tool-making, automotive, and aerospace industries, as well as in precision mechanics and many other areas of application.

Rotating parts, including shafts, are most commonly machined by turning. During this type of machining operation, the workpiece rotates at a certain angular velocity, which promotes vibration. The lower the contact stiffness of the workpiece, the greater its susceptibility to vibrations, which means that low rigidity shafts are particularly liable to chatter. The vibrations that occur during machining of shafts reduce the reliability of the turning process, affecting in a negative way the dimensional accuracy, waviness, and roughness of turned surfaces. Turning accuracy is commonly measured as the deviation *y* expressed in millimeters. If we assume that deviation is a function of *n*-arguments $y=f(x_{1},x_{2},\ldots ,x_{n})$, then controlling the turning process of low-rigidity shafts consists in minimizing the deviation by appropriately selecting explanatory variables.

### 1.1. The Problem of Low-Rigidity Shaft Machining

The traditional methods applied in the machining of rigid shafts are not effective when machining low-rigidity shafts. The effects of low rigidity of shafts can be mitigated using various solutions, such as multi-pass machining, machining at a reduced rotational speed, use of additional supporting elements, and even manual lapping [1]. These techniques, however, reduce the efficiency of production and increase the technical costs of manufacturing shafts. Besides, none of them guarantee the repeatability of the process parameters, and in many cases, the workpieces produced using these methods fail to meet accuracy requirements.

The approaches described in the literature for controlling the process of turning low-rigidity shafts are based on the assumption that the cutting tool moves relative to the workpiece in accordance with the kinematics of turning and vibrates in the direction of the pressure of the cutting force element and that the cutting edge imparts the profile of the tool bit tip onto the workpiece. Under the above assumptions, deformations of machined surfaces can be described using mathematical functions; however, an approach like this simplifies the problem under consideration. Utilitarian mathematical models completely ignore other factors (disruptions) that also have an impact on the machining process.

Factors that make it impossible to precisely predict the parameters of low-rigidity shaft turning can be divided into process-related, technological factors and workpiece-related, non-technological (operational) factors [2]. The technological factors are associated with the variability of the environment, variations in the cutting force, blunting of the tool bit, changes in the position of support reactions, the work-holding method used, contact deformations, and machining conditions and parameters (e.g., temperature, changes in voltage supplied to the lathe, type of coolant used).

Operational factors are related to the uncertainty associated with the fact that it is difficult to determine many properties, such as contact stiffness, which depends not only on the shape and initial state of the surface of the workpiece, its dimensions, and the type of material used but also on many other factors [3,4]. Due to the very large number of variables involved, the value of contact stiffness is also very difficult to determine. Other non-technological factors that can lead to the deformation of the surface of low-rigidity shafts during turning include: non-uniform workpiece material, dry and wet friction, fatigue strength, resistance to vibration and corrosion, as well as the type of treatment that the given part has previously been subjected to (e.g., heat treatment).

Mathematical models used to describe the turning of low-rigidity shafts ignore the disruptions arising during measurements and the fact that the turning tool bit does not vibrate solely in the direction of the cutting force vector. Other phenomena that mathematical models fail to capture include: the effect of the temperature of the part and the tool bit on the quality of the surface obtained during turning, the effect of wear of the tool bit tip on the reliability of turning, disruptions of turning kinematics expressed as deviations and anomalies in the way the tool bit moves relative to the workpiece.

### 1.2. Automation of the Low-Rigidity Shaft Turning Process

Process automation is one of the key megatrends that drive Industry 4.0 [5]. This concept relates to the use of the latest achievements in the fields of IT (Cloud Computing, Cloud Manufacturing, Internet of Things), cybernetics, mechatronics, and production engineering [6]. In particular, this applies to the integration and complete automation of control of production processes to the extent that decisions are made by machines, with humans playing a supervisory role. Because Industry 4.0 is a challenge and goal pursued by the world’s leading economies, there is large demand for research geared towards improving existing and developing new methods of process automation. 

Automation of solutions for the machining of high rigidity parts generally does not pose any major difficulties. The real challenge is the automation of machining of parts with atypical dimensional proportions, such as low-rigidity shafts [7]. When designing machine parts and the devices that are composed of those parts, it is necessary to take into account reliability, both in relation to the production process and the operation and maintenance. These two spheres, though distinct from a scientific point of view, complement each other in actual practice. To ensure high performance of a machine, it is necessary to use high quality parts. There are many factors that determine the quality of parts. They include, among others, high dimensional accuracy and low surface roughness. In order to produce high quality parts on an industrial scale, companies need to ensure that the production process meets high reliability standards, which can be achieved in conditions of high production automation. Ensuring a high level of automation in the production of low rigidity shafts is still a serious challenge, which is why there is a need for research aimed at developing effective and efficient methods of machining this type of machine parts.

### 1.3. The Application of Artificial Intelligence in the Machining of Parts

The literature provides numerous examples of application of artificial intelligence methods in controlling production processes. Most of them are solutions implemented in numerical machine tools. Yu, Kabaldin, and Shatagin, for example, describe the use of artificial neural networks for the classification of a point cloud in a 3D model of a workpiece, which allows to automatically analyze the shape of the workpiece machined in the working space of a CNC machine tool [8]. In turn, Moreira et al. applied fuzzy logic and neural networks to develop a controller for adjusting milling parameters in real time [9]. Mironova describes an intellectual system based on functional semantic network technologies, developed to ensure the accuracy of machining with point tools [10]. She found that axial misalignment of openings machined with high-speed steel drills depended on/was caused by tool advance and its rotational frequency. Sharma, Chawla, and Ram describe machine learning algorithms, namely support vector machine (SVM), restricted Boltzmann machine (RBM), and deep belief network (DBN), for the automatic programming of a computer numerical control machine [11]. Yusup et al. estimated optimal abrasive waterjet machining control parameters using artificial bee colony [12]. Fang, Pai, and Edwards developed a model for predicting roughness of machined surfaces. They used multilayer perceptron (MLP) neural networks to process multidimensional signals generated during metal machining operations, including three-dimensional cutting force signals and three-dimensional cutting vibration signals [13]. Naresh, Bose, and Rao report the results of a comparative study of artificial neural network (ANN) and adaptive neuro-fuzzy inference system models for the improved prediction of wire electro-discharge machining responses, such as material removal rate and surface roughness of a Nitinol alloy [14].

These examples indicate that machine learning methods can be used as effective predictive tools in controlling machining processes. Optimization methods can also be successfully used as components of production process control, as evidenced by various publications [6,15,16,17,18,19].

### 1.4. Innovative Aspects of the Proposed Approach

Increasing the level of automation of CNC machine tools can be obtained by the dynamic methods of machining control application. Yusuf Altintas presents many examples of open-loop and closed-loop robust nonlinear control systems in his book [20]. Modern control of such systems is based on adaptive methods such as feed drive control system, sliding mode controller with disturbance estimation, or intelligent machining module. With intelligent machining, tasks such as adaptive control, tool condition monitoring, and process control can be performed. Taking into account the diversity of tasks that are realized simultaneously during the modern automated CNC machining process, it must be mentioned that machining dynamics analysis methods can be used for the control of low-rigidity shafts turning process as supplemental ones. The combination of different control methods gives the possibility of reaching a synergy effect. However, such hybrid solutions need additional research in defined areas of CNC machining.

The chatter can be detected by continuously monitoring the amplitude of the sound spectrum. It can be measured by a microphone. Vibrations can be reduced by adjusting the cutting speed. However, it causes a reduction in the efficiency of the machining process.

Wang at al. [21] described an adaptive intelligent control system based on the constant cutting force and a smart machining tool. The smart cutting tool developed by the authors provided data on cutting force measurement, with a plug-and-produce feature, rendering a simple and compact low-cost sensing tool configuration. The authors state that the development of adaptive smart machining based on using smart cutting tools and the associated smart algorithms minimizes the machining time and improves surface roughness. The results presented in this paper completely prove this thesis.

The goal of this article is to present an original method for the adaptive control of turning low-rigidity shafts based on artificial intelligence and machine learning methods. A predictive controller algorithm was developed in which neural networks and genetic algorithms (GA) were implemented. The neural network generates the value of deviation *y* on the basis of two input variables: $F_{x1}$—moving tensile force and *e*—eccentricity of tensile force under tension. In mathematical terms, the neural network, treated as a black box, plays the role of an optimized fitness function given by the general Formula (1).
(1)$$y=\underset{F_{x1},e}{\min}\;f(F_{x1},e).$$

In the next step, the objective function (1) is optimized using a GA. The GA minimizes the deviation *y* by appropriately selecting force $F_{x1}$ and eccentricity *e*. The novelty of the presented concept lies in the proper training of the neural network, which, once it acquires the ability to generalize, can effectively convert input data into deviation. Owing to the fact that the neural network was trained using real-life data, the measurements take into account the impact of many interfering factors, which, in the case of machine learning, unlike in mathematical methods, are not known. Actually, all the necessary information is contained in the learning data set. The use of a neural network as an objective function for a GA is also a novel idea. By replacing the classic objective function, which is given by a detailed mathematical formula, with a neural network, we solved the problem of disruptions and difficult-to-define factors affecting the turning process. The level of process automation was also increased.

The article comprises four sections. Section 1 presents the theoretical aspects of turning low-rigidity shafts and a review of the relevant literature. Section 2 describes the key aspects of modelling the machining process and the application of algorithmic methods. Section 3 reports the results obtained using the algorithmic methods developed in this study and compares them to verify the effectiveness of the network methods used. The article concludes with Section 4, which contains observations and reflections made during the experiments, analyses, and modelling.

## 2. Materials and Methods

Figure 1 shows how the low-rigidity shaft was secured in the turning machine. The line defining the elastic limit of the workpiece is marked in red. There exist numerous theoretical methods for controlling the accuracy of machining elastically deformable shafts [22]. The accuracy of machining low-rigidity shafts can be effectively improved by increasing their stiffness via an oriented change in the elastic-deformable state [23]. This type of process control can be exerted by applying a tensile force to the workpiece, which, combined with the cutting force, produces longitudinal-transverse loads. Additionally, one can control the rotation angle of the cross-section of the workpiece at the holding point by applying a tensile force displaced relative to the axis of the lathe centers [24]. This work-holding solution can be depicted as a movable rotary support (Figure 1). The present experiments were carried out using this method of controlling the machining of low-rigidity shafts.

Formulas (2)–(4) specify the loading conditions of the shaft shown in Figure 1.
(2)$$\begin{array}{l} M_{1}=F_{x}\frac{d}{2},\\ M_{2}=F_{x1}\cdot e \end{array}$$
where: *d*—shaft diameter; *e*—eccentricity.
(3)$$y_{1}(x_{1})=\left(\frac{Q_{0}}{F_{x1}-F_{f}}\alpha \right)(\sinh\alpha_{1}x_{1})+\left(\frac{M_{0}}{F_{x1}-F_{f}}\right)(\cosh\alpha_{1}x_{1}-1),$$
where: $F_{f}$—axial component of cutting force.
(4)$$\begin{array}{cc} y_{2}(x_{2}) & =\frac{Q_{0}\cosh\alpha_{1}+M_{0}\alpha_{1}\sinh\alpha_{1}a+F_{b}e}{\alpha_{2}F_{x1}}(\sinh\alpha_{2}x_{2}-\alpha_{2}x_{2})\\ & +\left(\frac{\sinh\alpha \left(\frac{Q_{0}}{\alpha_{1}}\right)+M_{0}\sinh\alpha_{1}+M_{1}}{F_{x1}}\right)\left(\cosh\alpha_{2}x_{2}-1\right). \end{array}$$

Because the kinematic functions describing shaft turning do not take into account all the factors involved in the process, they are not sufficient to ensure an optimal level of control. In the present model, the turning process was optimized by minimizing the deviation function (where deviation is a measure of the quality of turning), which had a direct impact on the quality of the surface of machined shafts.

Figure 2 shows photographic images of the test stand used in the experiments. The spindle of the turning machine shown in the images is equipped with a rotary collet vise which allows to stretch the shaft secured in it. Apart from that, the position of the turning tailstock can be adjusted to change the angle of rotation of the shaft cross-section at the holding point by applying a tensile force displaced relative to the axis of the lathe centers. In this setup, the application of one controllable force factor (eccentric stretching) allows to produce two force factors in any pre-defined section of the specimen (particularly in the machining zone): longitudinal force *F_x_*_1_ and bending moment $M_{2}=F_{x1}\cdot e$, which counteracts cutting forces.

### 2.1. Data Preparation

Data for neural network training were collected during test stand experiments. A mechanical system was developed in which the process of machining a low rigidity shaft was controlled using two types of regulatory impacts—tensile force $F_{x1}$ and eccentricity *e*. An optimization problem described by the objective function (5) was formulated.
(5)$$y=\underset{F_{x1},e}{\min}(d.L,F_{f},v,a_{p},f,a,x,F_{x1},e)$$
where *d*—shaft diameter [mm]; *L*—shaft length [mm]; *F_f_* —axial component of cutting force [N]; *v*—cutting (feed) rate [mm/min]; *a*—distance from the cutting edge to the point at which the workpiece is secured in the spindle [mm]; *a_p_*—depth of cut [mm]; *f*—feed [mm/revolution].

It was assumed that all values of the parameters of the objective function (5), except for *F_x_*_1_ and *e*, remained constant during turning. A shaft with a length of L = 300 mm was subjected to turning. Figure 3, Figure 4, Figure 5 and Figure 6 show the curves of the objective function *y*, tensile force *F_x_*_1_, and eccentricity *e*, for predefined shaft diameters and cutting forces. Data collected during the tests were modelled using a two-dimensional gradient descent search algorithm. The section of the shaft from 1 to 300 mm was divided into 5981 0.05 mm long (parts corresponding to) measuring intervals. *F_x_*_10_ is the initial value of force *F_x_*_1_.

The data given above were used to train three types of neural networks: a shallow MLP ANN, a flat nonlinear autoregressive network with exogenous input (NARX) designed for the prediction of multidimensional time series and signals, and a long short-term memory (LSTM) neural network, which is a recurrent deep learning network. Algorithm 1 shows the workflow of the neuron-genetic controller.
**Algorithm 1** Two-dimensional gradient descent search algorithmdetermine the initial conditions by performing a turning operation to produce one low rigidity shaft:determine shaft diameter *d*determine bending force *F_be_*determine cutting force component *F_f_*determine the initial distance from the cutting edge to the point at which the workpiece is secured in the spindle *a_0_*determine initial axial force *F_x_*_10_determine initial eccentricity *e*_0_record measurements of *F_x_*_1_, *e* and deviation *y* every 0.05 mm by changing the values of parameters *F_x_*_1_ and *e* within the pre-defined rangeuse the measurement data to train an ANN to predict $y=f\left(F_{x1},e\right)$minimize deviation *y*, which is the output value of the neural network which serves as the objective function $y={\min\;}_{Fx1,e}\phi (d,L,F_{f},v,a_{p},f,a,x,F_{x1},e)$ for the GA

### 2.2. Shallow Neural Network

In the first variant of the experiment, an MLP ANN was developed (Figure 7). This network has three inputs: *a*, *F_x_*_1_, and *e*, one hidden layer containing 10 neurons and one output layer with a single *y* output representing deviation, which is a measure of the roughness of the machined shaft surface. A hyperbolic tangent sigmoid transfer function was used in the hidden layer, and a linear transfer function was used in the output layer.

Two measures of the quality of the trained network were used—mean square error (MSE) and regression *R*. Formula (6) gives the method of calculating MSE:(6)$$MSE=\frac{1}{n}\sum_{i=1}^{n}{\left(y_{i}^{\prime }-y_{i}^{*}\right)}^{2}$$
where *n*—number of cases in a given set; $y_{i}^{\prime }$—reference value for the *i-th* shaft section; $y_{i}^{*}$—predicted value for the *i-th* shaft section.

The method of calculating the regression coefficient *R* is given by Formula (2):(7)$$R(y^{\prime },y^{*})=\frac{\mathrm{cov}(y^{\prime },y^{*})}{\sigma_{y^{\prime }}\sigma_{y^{*}}}\;R\in \langle 0,1\rangle$$
where $\sigma_{y^{\prime }}$—standard deviation of reference values, $\sigma_{y*}$—standard deviation of predicted values.

In Table 1, the data set with 5981 cases is divided into three subsets: a training subset, a validation subset, and a test subset in a ratio of 70:15:15. Table 1 also shows MSE and R values determined for variant (I) data, which are visualized in Figure 3. 

The generalization capacity of a trained neural network is better the lower the value of MSE and the higher the value of R. Figure 8 shows a plot of MSE over training epochs. The learning curve (Figure 8a) has a regular hyperbolic shape, which testifies to the high quality of the trained network. The high degree of overlap between the curves for the training, validation and test subsets also confirms that the network has a high ability to generalize predictions and that there is no overfitting.

To prevent overfitting of the ANN, the early stopping technique was used. The method consists in monitoring the validation error in the individual epochs. If the error did not decrease for six consecutive epochs, training was terminated. Network training was also constrained by setting a limit on the maximum number of epochs. In the case under consideration, this limit was 20 epochs. The ANN was trained using the Levenberg–Marquardt (LMA) optimization algorithm, which includes finding the zeros of Newton’s function. This type of algorithm, also known as a back error propagation algorithm, is characterized by high speed and high memory requirements. Two important parameters of LMA are gradient and momentum (Mu). Figure 9b shows a graph of gradient values during ANN training. The lack of clear fluctuations and the downward trend demonstrate that the training procedure worked well.

Figure 9c shows a graph of Mu values. The momentum decreases and is analogous to the inertia of the search for the minimum point of the objective function; therefore, the closer to the minimum sought, the lower the Mu. Figure 9a shows an error histogram. The fact that the shape of the histogram resembles a normal distribution curve and that the largest number of errors has the lowest values demonstrates that the trained network is good quality and that there are no symptoms of overfitting.

### 2.3. NARX Neural Network

In the second variant of the experiment, we used a NARX with feedback connections. NARX are recurrent dynamic neural networks designed to predict single or multiple time series. Prediction can be made in the so-called closed-loop model, which means that the output values are passed back to the input, thus supporting the prediction. In the case at hand, NARX network inputs included components *F_x_*_1_ and *e*, and also, in the variant with a closed-loop NARX, the previously obtained actual value of deviation *y_t-_*_1_. The equation that defines the operation of the NARX network is given by Formula (8) [25]:(8)$$y(t+1)=F(y(t),y(t-1),y(t-2),\ldots ,y(t-n_{y}),x(t+1)x(t),x(t-1),x(t-2),\ldots ,x(t-n_{x}))$$
where F(⋅)—mapping function; y(t−1)—output of NARX at moment *t* for moment *t* + 1; $y(t),y(t-1),\ldots ,y(t-n_{y})$—actual past values of the signal; $\left(t+1)x(t),x(t-1),\ldots ,x(t-n_{x}\right)$—sequential values of the NARX input signal; $n_{y}$—number of outputs; $n_{x}$—number of inputs. According to Formula (8), the next value of output signal y(t) is regressed on previous values of the input signal and previous values of the output signal.

Figure 10 shows the structure of the NARX network. The network has two inputs, which are signals with values *F_x_*_1_ and *e*. The hidden layer contains 10 neurons, and the output layer consists of a single deviation signal *y*. The network was created and trained in open-loop form as shown in Figure 10a. Open loop (single-step) training is more efficient than closed loop (multi-step) training. An open loop allows the network to be fed with the correct previous output values to produce the correct current outputs. After training, the network is converted to a closed-loop form required by the application.

Table 2 shows data divided into training, validation, and test subsets, as well as the results of training the NARX open-loop network. It is worth noting that although the training results are better than for the shallow ANN, they are not final results yet. The actual quality of a NARX network is measured by calculating MSE and R parameters after it has been transformed into a closed-loop network.

Figure 11a,b confirm the high quality of NARX training. The MSE curves for the training, validation, and test subsets are almost identical. Training was terminated after 20 epochs.

Similar to ANN, early stopping was used to protect the NARX network against overfitting. The LMA algorithm was used to train the NARX network.

Figure 12a–c also confirms the high quality of the training process. The explanations and conclusions that can be drawn from the analysis of the data shown in those figures are analogous to those for Figure 9a–c.

Table 3 shows the results obtained after the open-loop NARX network had been converted into a closed-loop network. It can be seen that the NARX network which predicts results a step ahead, shows excellent performance. The quality of the prediction for the whole sequence is much worse; however, this was not important in the context of the present study, because predictions were not made here for horizons longer than one step. For this reason, indicators of quality of closed-loop NARX for whole sequence prediction were not taken into account when assessing the performance of the neural networks for controlling the process of turning low-rigidity shafts.

Figure 13 shows regression statistics for the closed-loop step-ahead NARX for all cases from the training, validation, and test subsets. Figure 13a shows regression for the entire set, which is close to 1. Figure 13b shows data for 16 randomly selected cases, allowing to observe deviations of predictions from the reference value (target). For this set of 16 cases, R = 0.99946, which is confirmed by the high degree of overlap of measuring points and a fit line with the ideal prediction line Y = T.

### 2.4. Deep Network LSTM

In the third variant of the predictive model for controlling the process of turning low-rigidity shafts, we used a deep LSTM neural network. LSTM is a recurrent network. It has a more complex structure than MLP, which endows it with special properties for learning long-term relationships between individual sequential cases. Figure 14 shows the workflow of the LSTM network [26].

Each of the LSTM layers contains two states, where *h_t_* is the hidden (initial) state at moment t and *c_t_* is the cell state at moment t. The cell state contains information learned in previous time steps. At each stage, each LSTM layer adds or removes information from the cell state. Information is updated using gates. The gates have the task of controlling the level of cell state: f—reset (forget), (i)—the input gate (update) controls the level of cell state update, g—candidate cell (update), (o)—output gate.

Equation (9) describe the components of the LSTM layer at time step *t*:
(9)$$\begin{array}{c} f_{t}=\sigma_{g}(W_{f}x_{t}+R_{f}h_{t-1}+b_{f})\\ g_{t}=\sigma_{c}(W_{g}x_{t}+R_{g}h_{t-1}+b_{g})\\ i_{t}=\sigma_{g}(W_{i}x_{t}+R_{i}h_{t-1}+b_{i})\\ o_{t}=\sigma_{g}(W_{o}x_{t}+R_{o}h_{t-1}+b_{o}) \end{array}$$
where *W*—weights, *R*—recurrent weights, *b*—biases, *σ*—sigmoidal gate activation functions expressed by $\sigma (x)={\left(1+e^{-x}\right)}^{-1}$, $h_{t}$—hidden state at time step *t* described as $h_{t}=o_{t}\circ \sigma_{c}(c_{t})$, where *σ_c_* is the state activation function. The cell state at a given time step *t* is described by $c_{t}=f_{t}\circ c_{t-1}+i_{t}\circ g_{t}$ where ∘ denotes element-wise multiplication of vectors. 

The architecture of the LSTM network used in this study is shown in Table 4. As in the NARX network, the input layer of the LSTM network consists of variables *F_x_*_1_*, e* and the *y* value imported from the input. Accordingly, there are three activations in the input layer. The second layer is a bidirectional LSTM layer (BiLSTM) with 200 activations. It learns bidirectional long-term dependencies between steps of sequences. Such dependencies can be useful when the network should learn from full time series at each stage. The next layer is a fully connected layer with one activation. The last layer is the regression output variable *y*.

The LSTM network was trained using the adaptive moment estimation optimization method (ADAM), for which: regularization factor L2 = 1 × 10^−4^, initial learning rate 0.05, learn rate drop factor 0.1, learning period 10, momentum 0.9. The learning conditions included a maximum of 5 epochs and a minimum batch size of 64. Two measures of learning quality were used—RMSE and loss. RMSE is the rooted MSE (10).
(10)$$RMSE=\sqrt{\frac{1}{n}\sum_{i=1}^{n}{\left(y_{i}^{\prime }-y_{i}^{*}\right)}^{2}}$$

The loss function is given by Equation (11):
(11)$$Loss=-\sum_{i=1}^{n}y_{i}^{\prime }\log(y_{i}^{*})/m$$
where *m*—number of observations, *n*—number of responses, $y_{n}^{\prime }$—reference values, $y_{n}^{*}$—response values.

The learning effectiveness of LSTM is illustrated in Figure 15 and Figure 16. The RMSE and loss curves are similar and they both show that the learning process proceeded in a correct manner. The initially high error values and losses decreased quickly to eventually stabilize at a constant level. Then, network training was terminated.

Table 5 presents additional parameters illustrating the learning process. Mini-batch RMSE stabilized after the first epoch, while mini-batch loss stabilized after the second epoch. The base learning rate was 0.05 throughout the entire learning process.

Table 6 presents the results of training the LSTM network. Step-ahead prediction was very effective, although the MSE parameter was slightly lower than in the case of NARX. Whole sequence prediction was much less effective. This, however, was of no consequence for the present experiments, because this type of prediction was not used in controlling the accuracy of the machining of shafts.

### 2.5. GA-Based Controller

GA are based on natural evolutionary processes. In nature, individuals that are best adapted to specific conditions have the best chances of survival and reproduction. As a result, subsequent generations are even better adapted than the previous ones, because they have inherited the best traits (ones that are best suited to their living conditions) from their parents. The same idea is used in evolutionary computational algorithms. GA are able to solve optimization problems with both real and integer types of constraints. They are based on a stochastic, population algorithm that searches randomly by mutation and crossover among elements of the population. Each population consists of a set of chromosomes, and each chromosome is a vector composed of genes. Genes have binary values of 0 or 1. The computational process for a classical GA comprises six stages: encoding, evaluation, selection, crossover (reproduction), mutation, and decoding. 

Encoding consists in stochastically generating the initial population. In the next step, the degree of fit of each chromosome is evaluated by calculating the fitness function value for each chromosome. In the present case, a neural network plays the role of fitness function. The higher the value of the objective function for a given chromosome, the better suited it is to solve the problem described by the objective function. The evaluation parameters assigned to chromosomes determine the likelihood that a given chromosome will be carried to the next stage (mutation). Mutation is the transformation $O_{m}:D(P)\to D(P)$ which randomly alters the *l*-th component of the solution (chromosome) $X_{i}^{t}$ at a predefined probability: $O_{m}(X_{i}^{t})=X_{i}^{t+1}$, where: $X_{i}^{t}=(x_{1},\ldots ,x_{i},\ldots ,x_{n}),$
$X_{i}^{t+1}=(x_{1},\ldots ,\bar{x_{i}},\ldots ,x_{n})$. Crossover *O_k_* is the transformation $O_{k}:D(P)\times D(P)\to D(P)\times D(P)$, where $X_{i}^{t}=(x_{1},\ldots ,x_{n})$,$X_{i}^{t+1}=(x_{1},\ldots ,x_{i},v_{i+1},\ldots ,v_{n})$ and $X_{j}^{t}=(v_{1},\ldots ,v_{n}),$$X_{i}^{t+1}=(v_{1},\ldots ,v_{i},x_{i+1},\ldots ,x_{n}).$
Figure 17 shows a general scheme of the neural-genetic controller.

A genetic minimizer was used to control the machining of low-rigidity shafts. The optimization problem can be formulated as $\min_{x}f(X),$ where *X*—vector of input variables. Figure 18 shows the best fitness plot of the GA. It plots the best function value in each generation compared to the iteration number. In the present optimization, the best fitness function value was 0.35655, and the mean value was 0.356526.

## 3. Results and Discussion

As previously mentioned, the main task of the predictive controller algorithm is to minimize the deviation function (1). The deviation *y* is directly correlated with the surface roughness of a machined shaft, and it is a measure of the quality of turning. The quality of the neural network prediction is expressed by means of the MSE and R indicators (Table 7). The more accurate the prediction of the neural network and the higher the level of optimization of the fitness function performed by the genetic algorithm, the smaller the deviation *y* that expresses the quality of turning.

In order to best assess the quality of the neural networks used, a number of cases were extracted from the training subset before training, which were later used to test the individual network variants. Table 7 shows the results of tests determining the performance of the individual types of neural networks in controlling the machining of low-rigidity shafts.

An analysis of the data given in Table 7 indicates that the best results for the tested data set were obtained using the LSTM network. It is worth noting that the differences in MSE and R between the three types of networks are negligible. When MSE and regression R are considered, even the results of the least effective network (MLP ANN) are sufficient to control the process of turning low-rigidity shafts. LSTM and NARX are all the more suitable for this purpose.

### 3.1. Shallow MLP Network

An MLP ANN is fundamentally different from a NARX and LSTM. It not only has a distinct structure and lacks feedback and recurrent solutions, but, above all, it does not take into account the order of occurrence of the individual measurements on the time axis. For this reason, ANNs are rarely used to predict time sequences or event sequences. This does not mean, however, that they cannot be used for those purposes.

Parametric data collected during the turning of shafts constitute a certain sequence. To provide sequencing information, the value of parameter *a* (distance from the cutting edge to the point at which the workpiece is secured in the spindle) was entered into the MLP ANN input data vector (Figure 1). As a result, each three-component MLP ANN input vector, consisting of variables *a*, *F_x1_*, and *e*, now had a specific index (variable *a*). This allowed to extract the test subset from the 5981-element data set, by first randomly mixing the cases, and then cutting off cases 1 to 5500 for the training set and cases 5501 to 5981 for the test set. The test subset obtained in this way contained 481 cases. This means that ANN was trained on a set which was unordered but indexed by the value of *a*. Owing to this, training produced very good results. Figure 19 shows the differences between predicted values and reference values of deviation *y*. Figure 20 shows a detail of Figure 19 to better visualize the deviations between the prediction line and the reference line. The 481 measurements were plotted on the horizontal axis in such a way that this axis corresponded to the total length of the machined shaft L = 300 mm.

### 3.2. NARX Neural Network

In the case of NARX, it was impossible to apply the method of extraction of the test set used for ANN. This was because NARX had no index in the input vector. To preserve the order of the sequence, NARX was designed to include feedback connections, where the previous input value of deviation *y_-t_*
_1_ was provided as a third input vector component, beside *F_x_*_1_ and *e*. Figure 21 shows the deviation of predicted values from reference values for the NARX network. As in the case of ANN, 481 measurements were plotted on the horizontal axis so that this axis corresponded to the total length of the machined shaft L = 300 mm. Figure 22 shows the machining quality prediction (a detail of Figure 21) to better visualize the deviations between the prediction line and the reference line.

### 3.3. Deep LSTM Network

The data for training LSTM were prepared in the same way as for NARX. The input vector was also the same for both networks. A comparison of the deviations of the LSTM network shown in Figure 23 and Figure 24 with the deviations of the NARX network show very large similarities in prediction. This is associated with the similar nature of the two networks, which, due to the use of feedback *y_t-1_*, are well suited for predicting time series and sequences and therefore can be employed for predicting various types of processes.

### 3.4. Neural-Genetic Controller

Figure 25 and Figure 26 show examples of the results of applying the GA-based optimizer, in which a neural network took over the role of the objective function. Figure 25 shows two cases for L = 300 mm in which the controller adjusts the parameters e and *F_x_*_1_ to minimize deviation *y*. In the first case, optimization was performed for turning length a = 100 mm and in the second case for a = 200 mm. As can be seen, there is a space between the two curves that allows to select parameters e and *F_x_*_1_ within a certain range. The complex shape of the curves implies that the relationship between the two parameters is characterized by a high level of complexity. There are also big differences in the course of both curves, especially in the F_x1_ range from 800 N to 1050 N. 

Figure 25 shows a plot that is similar to that in Figure 24, but this time for a = 250 mm.

In Figure 25 and Figure 26, there are many optimal pairs of parameters *e* and *F_x_*_1_ for particular values of turning length *a*. As a consequence, it is possible to fix one of these parameters for a defined constant value that is not modified during the machining. At the same time, the second value can be adjusted. It can happen that the controller issues a command to fix the value of parameter *e* or *F_x_*_1_ beyond the permissible range. In such cases, both of the parameters must be changed simultaneously.

The tests showed that the quality of GA-based control of the turning process mainly depends on the effectiveness of the objective function. Therefore, it can be assumed that the most effective variant among the ones investigated in this study is the one that combines LSTM with GA. 

## 4. Conclusions

This article presents an original approach to controlling the process of turning low-rigidity shafts with the use of a hybrid neural-genetic controller. It was assumed that the use of an ANN in place of a GA objective function would increase the effectiveness of control compared to other known methods. Direct comparisons with other methods of controlling the machining of this type of shafts are not possible without maintaining exactly the same material, machine and measurement conditions, and parameters. For this reason, we compared three variants of machine learning algorithms we developed especially for this study: MLP ANN, NARX, and LSTM. 

The tests confirmed that properly prepared measurement data are of key importance for the quality of the controller. A prerequisite for high-quality prediction, and thus for effective optimization and ultimately control, is the use of a training set that includes measurements for the full range of turning lengths. Hence, before starting the production of a new batch of products, a pilot process of turning one reference shaft should be performed. This step is necessary for the acquisition of training data. This study shows that improper division of measurement data into training and test subsets may seriously reduce the quality of prediction and, consequently, the efficiency of control.

Because the best performing controllers used feedback, their performance might constitute a crucial utilitarian problem. If the controller works too slowly, then either sampling has to be done less frequently or the turning process must be slowed down. It should be stated that in the present case of controlling the turning process, the quantity of data did not cause any efficiency problems. The neural networks were trained in a few seconds, and the results were generated in an even shorter time. One limitation of the proposed solution is that the GA, being iterative, slowed down the optimization, but it can be replaced by other, faster optimizers. It is all a matter of give and take between the speed and effectiveness of optimization.

A clear advantage of the presented solution is that it allows to bring to light and take into account many invisible but important factors that affect the effectiveness of control. Real-life data contain information that, for obvious reasons of space, cannot be included in mathematical models. Although all models, including both mathematical and neural models, constitute a simplified representation of real objects, neural networks can reproduce real-life processes more accurately because they have the ability to generalize and take into account large amounts of information contained in measurement data.

## Figures and Tables

**Figure 1 sensors-20-04683-f001:**
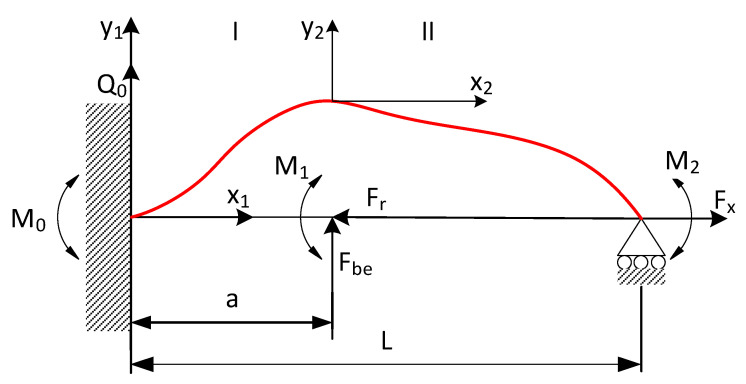
Work-holding method for securing the low-rigidity shaft specimen in the turning machine. Notation: *F_be_*—bending force exerted by the cutting tool bit, *F_x_*—tensile force along the x axis. *x_2_, y_1_, y_2_*—current coordinates at each section of the workpiece, *a*—distance from spindle to the tip of the cutting tool bit, *L*—length of shaft, *M_0_, Q_0_*—initial parameters: moment and transverse force at the holding point, respectively, *M_1_*—moment generated by the axial component of cutting force, *M_2_*—moment generated at the holding point at which the part is secured to the tailstock of the turning machine.

**Figure 2 sensors-20-04683-f002:**
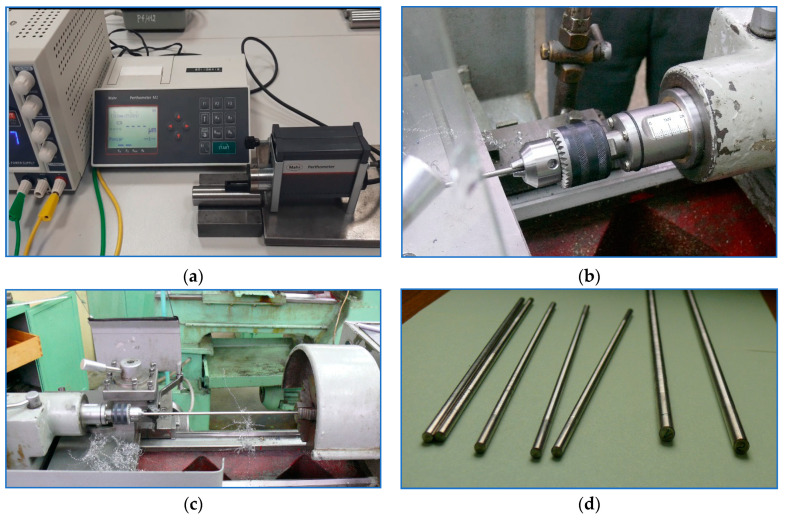
(**a**) Roughness measuring instrument; tailstock collet assembly for machining elastic-deformable shafts: (**b**) idle position, tensile force of 2 kN; (**c**) view of the test stand with the shaft secured in the lathe (Ø6, L = 300 mm); (**d**) specimens.

**Figure 3 sensors-20-04683-f003:**
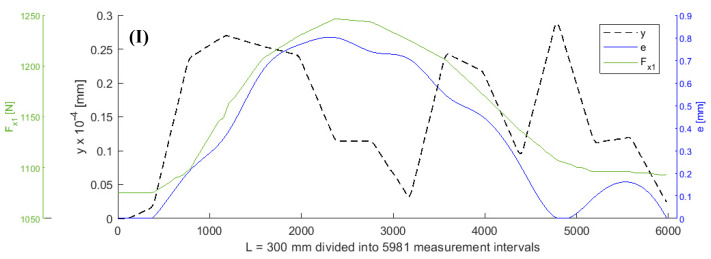
Curves of objective function y, tensile force *Fx_1_*_,_ and eccentricity *e* for *d* = 6 mm, *F_be_* = 49 N, *Fx_1_* = 980 N, *L* = 300 mm, *F_f_* = 30 N.

**Figure 4 sensors-20-04683-f004:**
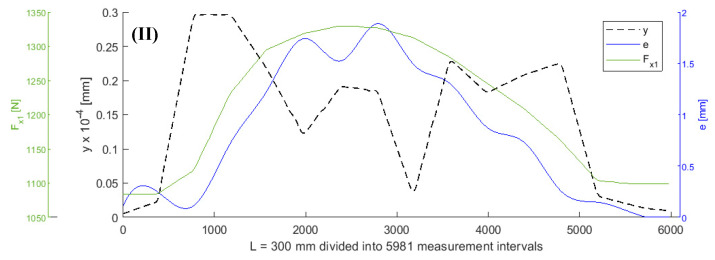
Curves of objective function y, tensile force *Fx_1_*, and eccentricity *e* for *d* = 6 mm, *F_be_* = 70 N, *F_x10_* = 980 N, *L* = 300 mm, *F_f_* = 40 N.

**Figure 5 sensors-20-04683-f005:**
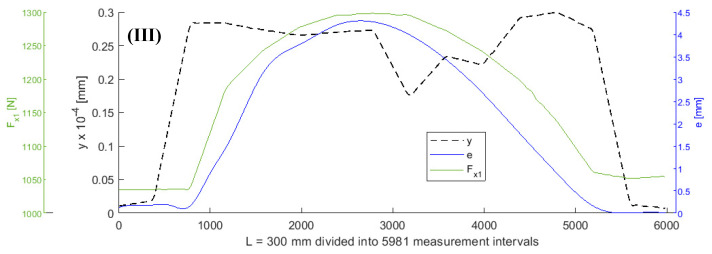
Curves of objective function y, tensile force *Fx_1_*, and eccentric *e* for *d* = 8 mm, *F_be_* = 147 N, *Fx_10_* = 980 N, *L* = 300 mm, *F_f_* = 196 N.

**Figure 6 sensors-20-04683-f006:**
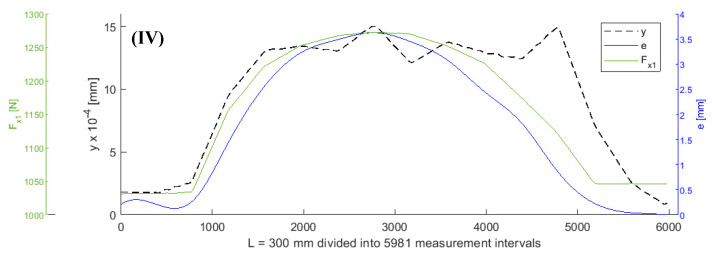
Curves of objective function y, tensile force *Fx_1_*, and eccentric *e* for *d* = 8 mm, *F_be_* = 147 N, *Fx_10_* = 980 N, *L* = 300 mm, *F_f_* = 196 N.

**Figure 7 sensors-20-04683-f007:**
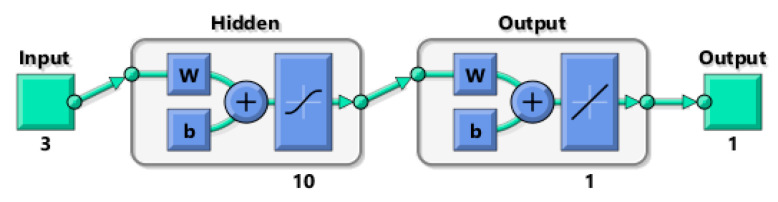
Structure of the shallow neural network.

**Figure 8 sensors-20-04683-f008:**
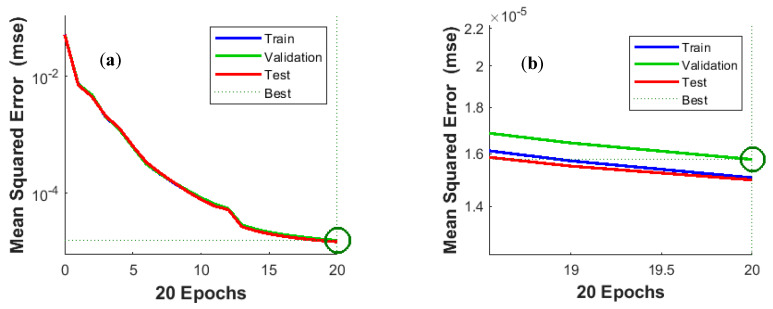
Best validation performance is 1.5775 × 10^−5^ at epoch 20: (**a**) general view, (**b**) enlarged view of the terminal part of the curve.

**Figure 9 sensors-20-04683-f009:**
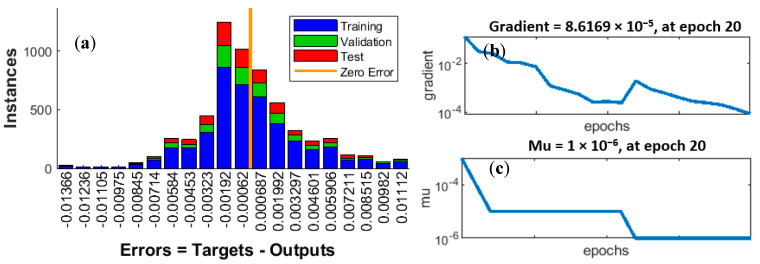
(**a**) Error histogram with 20 bins, (**b**) gradient curve, (**c**) Mu curve.

**Figure 10 sensors-20-04683-f010:**
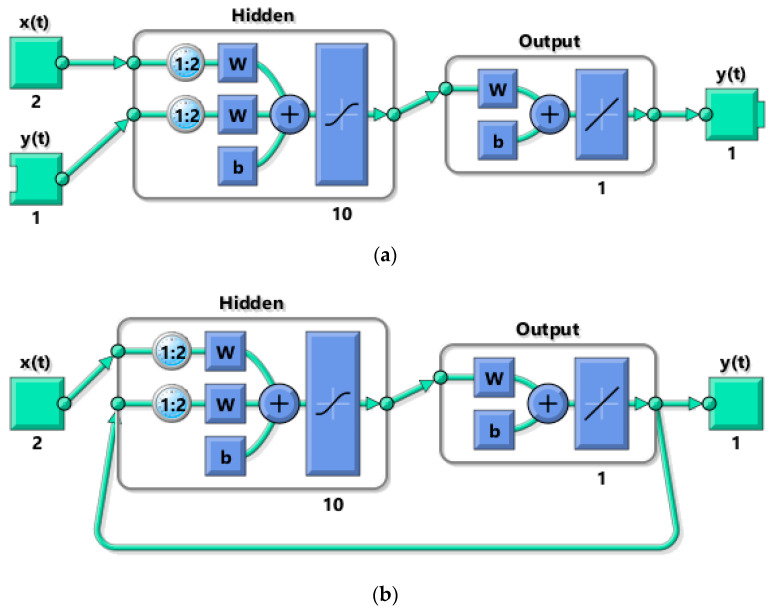
Structure of nonlinear autoregressive network with exogenous input (NARX) neural network: (**a**) open-loop architecture; (**b**) closed-loop architecture.

**Figure 11 sensors-20-04683-f011:**
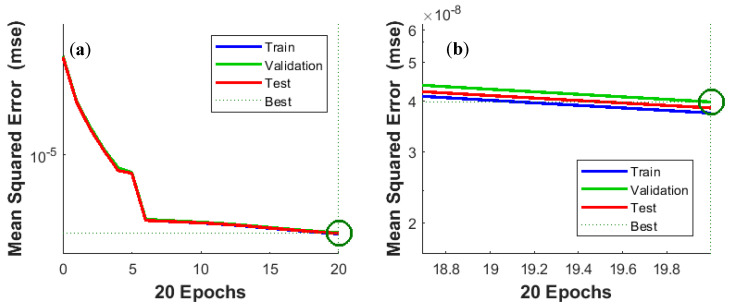
Best validation performance is 3.9897 × 10^−8^ at epoch 20: (**a**) general view, (**b**) enlarged view of the terminal part of the curve.

**Figure 12 sensors-20-04683-f012:**
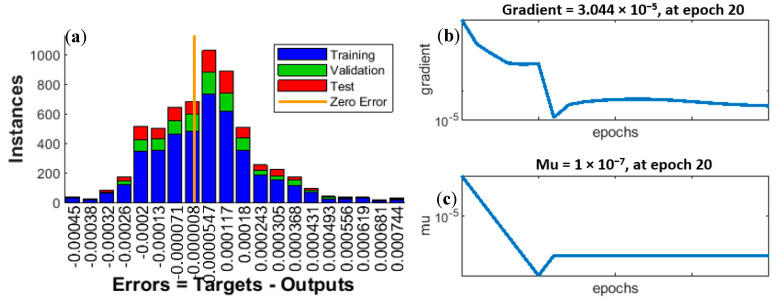
(**a**) Error histogram with 20 bins, (**b**) gradient curve, (**c**) Mu curve.

**Figure 13 sensors-20-04683-f013:**
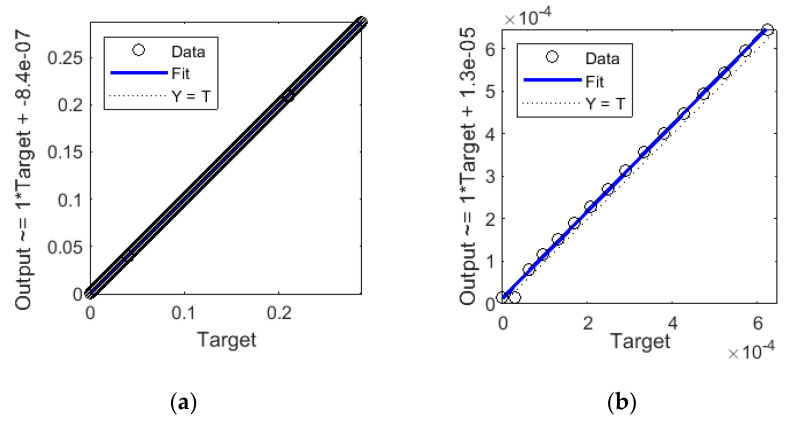
Regression statistics for closed-loop step-ahead NARX: (**a**) R ≈ 1 for whole set of 5980 cases, (**b**) R = 0.99946 for the subset of 16 cases.

**Figure 14 sensors-20-04683-f014:**
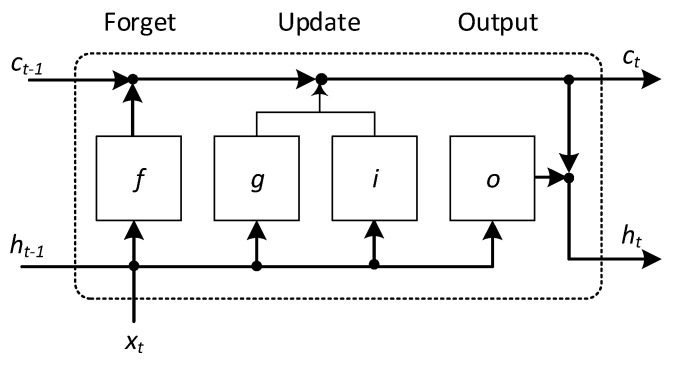
Structure of a long short-term memory (LSTM) layer [26].

**Figure 15 sensors-20-04683-f015:**
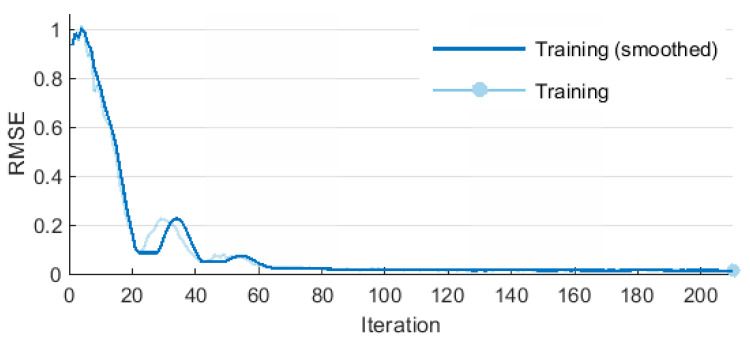
Training performance for LSTM.

**Figure 16 sensors-20-04683-f016:**
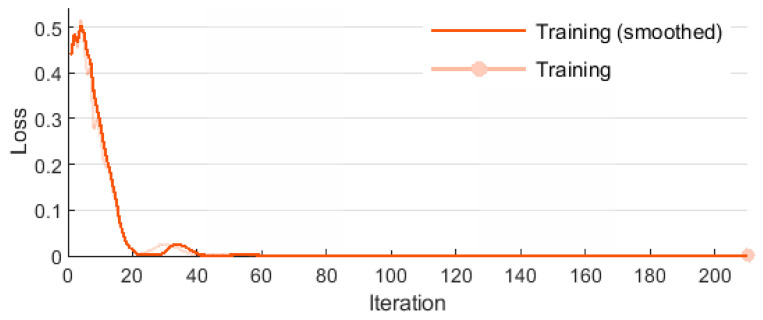
Training loss for LSTM.

**Figure 17 sensors-20-04683-f017:**
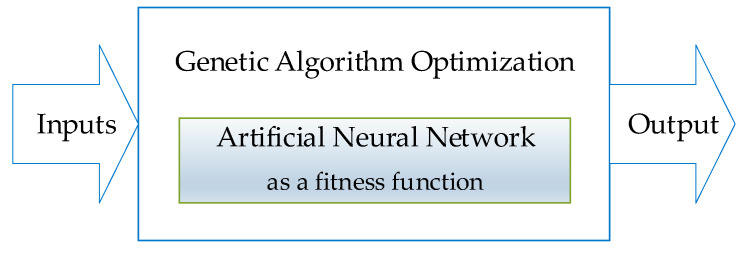
Neural-genetic controller.

**Figure 18 sensors-20-04683-f018:**
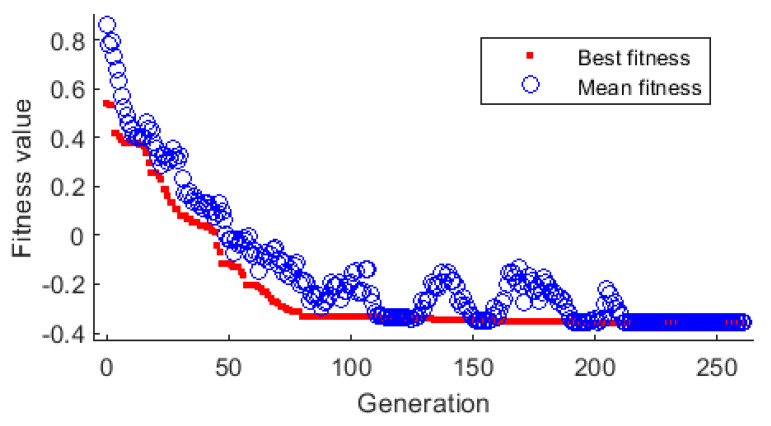
Genetic algorithm—the best fitness plot.

**Figure 19 sensors-20-04683-f019:**
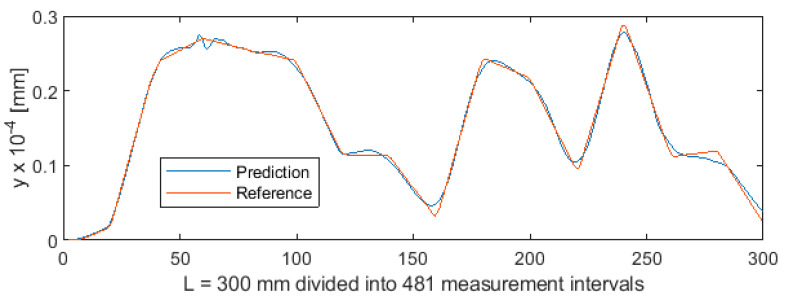
Machining quality prediction using MLP ANN.

**Figure 20 sensors-20-04683-f020:**
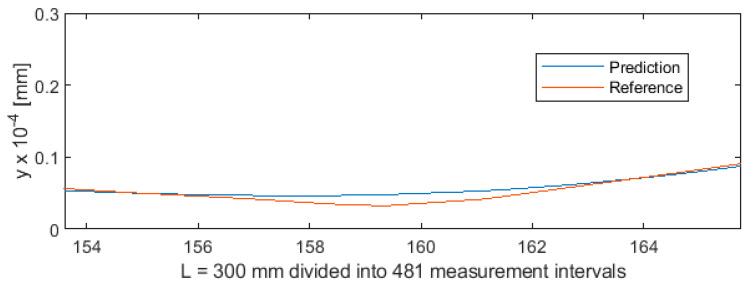
Machining quality prediction using MLP ANN—detail of the process for L = 154 ÷ 165 mm in Figure 19.

**Figure 21 sensors-20-04683-f021:**
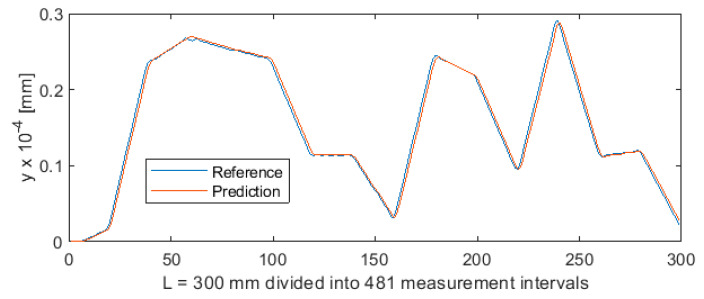
Machining quality prediction using NARX.

**Figure 22 sensors-20-04683-f022:**
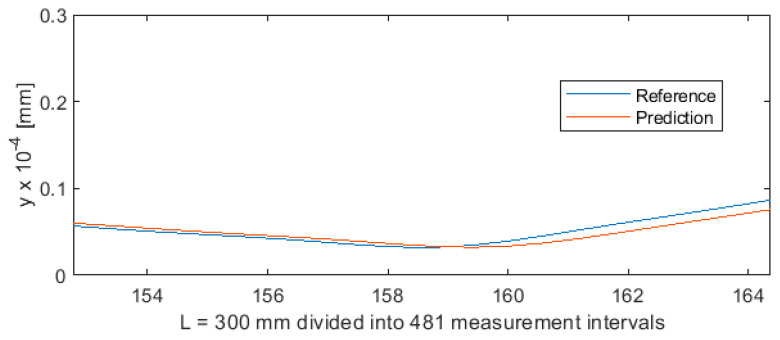
Machining quality prediction using NARX—detail of the process for L = 154 ÷ 165 mm in Figure 21.

**Figure 23 sensors-20-04683-f023:**
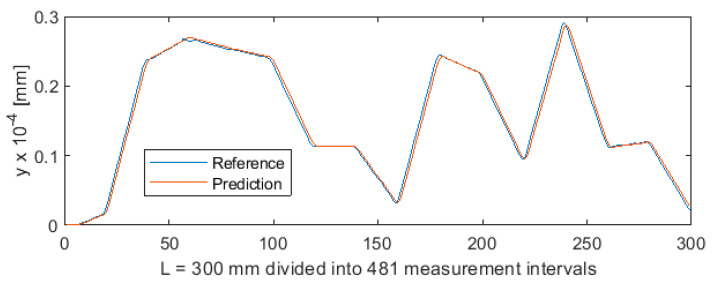
Machining quality prediction using LSTM.

**Figure 24 sensors-20-04683-f024:**
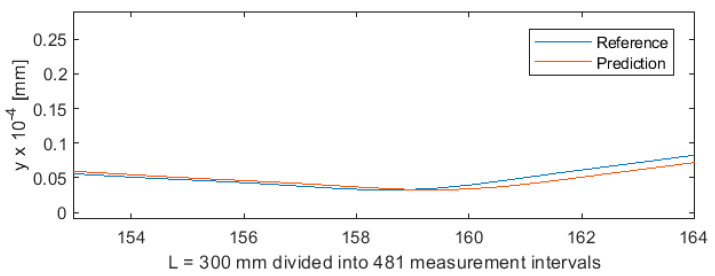
Machining quality prediction using LSTM—detail of the process for L = 154 ÷ 165 mm in Figure 22.

**Figure 25 sensors-20-04683-f025:**
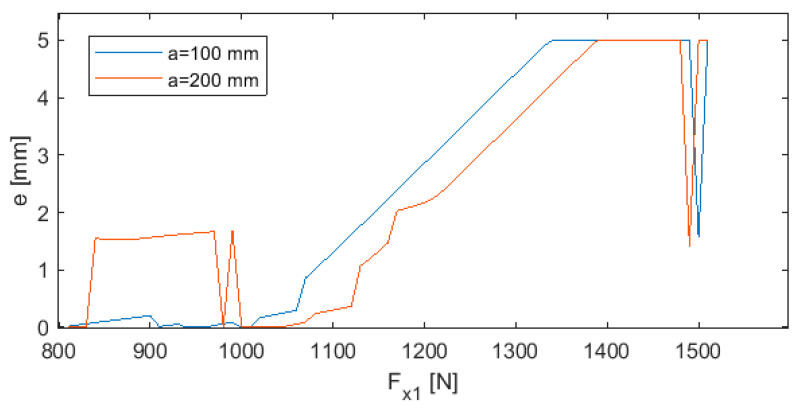
Neural-genetic controller for a = 100 mm and a = 200 mm.

**Figure 26 sensors-20-04683-f026:**
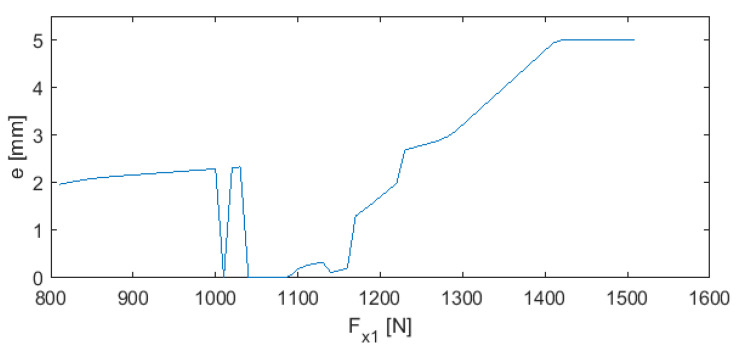
Neural-genetic controller for a = 250 mm.

**Table 1 sensors-20-04683-t001:** Training results for the multilayer perceptron (MLP) artificial neural network (ANN) by data subset.

Data Subset	Number of Cases in Set	Mean Square Error (MSE)	Regression (R)
Training set (70%)	4187	1.5059 × 10^−5^	0.99886
Validation set (15%)	897	1.5775 × 10^−5^	0.99880
Testing set (15%)	897	1.4976 × 10^−5^	0.99878

**Table 2 sensors-20-04683-t002:** Training results for the open-loop NARX by data subset.

Data Subset	Number of Cases in Set	Mean Square Error (MSE)	Regression (R)
Training set (70%)	4187	3.7450 × 10^−8^	0.999
Validation set (15%)	897	3.9897 × 10^−8^	0.999
Testing set (15%)	897	3.8548 × 10^−8^	0.999

**Table 3 sensors-20-04683-t003:** Closed-loop NARX training results.

Closed-Loop NARX	Mean Square Error (MSE)	Regression (R)
step-ahead prediction	3.7982 × 10^−8^	0.9999
whole sequence prediction	9.7246 × 10^−3^	0.5506

**Table 4 sensors-20-04683-t004:** Layers of the LSTM after feature extraction.

#	Layer Description	Activations	Learnable Parameters (Weights and Biases)
1	Sequence input with 3 dimensions	3	–
2	BiLSTM with 200 hidden units	200	Input weights: 800 × 2; Recurrent Weights: 800 × 200; Bias: 800 × 1.
3	One fully connected layer	1	Weights: 6 × 200; Bias: 1 × 1.
4	Regression output	–	–

**Table 5 sensors-20-04683-t005:** Layers of the LSTM after feature extraction.

Epoch	Iteration	RMSE Mini-Batch	Mini-Batch Loss	Base Learning Rate
1	1	1.06	0.6	0.05
1	50	0.07	2.1 × 10^−3^	0.05
2	100	0.01	1.0 × 10^−4^	0.05
2	150	0.01	8.9 × 10^−5^	0.05
3	200	0.01	7.9 × 10^−5^	0.05
3	250	0.01	1.1 × 10^−4^	0.05
4	300	0.01	8.7 × 10^−5^	0.05
4	350	0.01	8.7 × 10^−5^	0.05
5	400	0.02	1.1 × 10^−4^	0.05
5	450	0.01	7.0 × 10^−5^	0.05

**Table 6 sensors-20-04683-t006:** Training results for the closed-loop LSTM.

Closed-Loop LSTM	Mean Square Error (MSE)	Regression (R)
step-ahead prediction	1.4067 × 10^−4^	0.9999
whole sequence prediction	2.6045 × 10^−2^	0.5506

**Table 7 sensors-20-04683-t007:** The results of neural network tests.

Neural Network Type	MSE	R
Deep LSTM (step-ahead prediction)	1.5456 × 10^−5^	0.9999
Shallow MLP ANN	2.3984 × 10^−4^	0.9997
NARX (step-ahead prediction)	1.8819 × 10^−5^	0.9999

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
