# Peer review of "The Use of Neural Networks and Genetic Algorithms to Control Low Rigidity Shafts Machining"

_sensors, 2020, doi:10.3390/s20174683_

Round 1
Reviewer 1 Report
This paper presents some interesting research work on developing the machine-learning-based automated approach for control of the machining process for low-rigidity shafts and its implementation perspectives. The work has some interesting ideas supported with some experimental trial data. However, the paper needs to undertake the following revisions in order to reach the acceptable level for its publication at the journal:
- The paper should include a ‘Nomenclature’ to list all symbols and abbreviations used in the paper manuscript.
- Section 3 is better titled ‘Results and Discussion’, while Section 4 should be titled ‘Conclusions’.
- Section 5 Patents is unnecessary for a research paper, some of those patents can be listed in References section if research wise needed. Please notice that some patents are more applied, rather than suitable for being included in a journal publication.
- In Section 2.1, the paper claims ‘A mechanical system was developed in which the process of machining a low rigidity shaft was controlled using two types of regulatory impacts – tensile force Fx1 and eccentricity e’. However, machining instability and chatter are often occurred in machining low rigidity shafts, which are analysed by machining dynamics analysis methods. It is strongly suggested the paper should include a paragraph to elaborate the innovative aspects of the proposed approach and discuss how and whether the machining dynamics analysis will be applicable to the approach.
- Figures 18-23 illustrate the ‘machining quality prediction…’, the concept of machining quality is too vague. Is it referred to the machine surface roughness? This should be professionally specified and clarified on its modelling and analysis, which is essentially important for the later-on computational prediction.
- Furthermore, the proposed approach seems being unable to measure the cutting force or tensile force Fx1 in-process and how the approach can adaptively or even smartly control the process against the dynamic variation of the cutting dynamics? This should be further discussed in the paper appropriately.
- The following very relevant books and papers in the field should be included in References particularly against comments (4), (5) and (6):
- Y. Altintas, Manufacturing Automation: Metal Cutting Mechanics, Machine Tool Vibrations, and CNC Design. 2nd Edition, 2012, Cambridge University Press.
- C. Wang, S.B.C. Ghani, K. Cheng and R. Rawkoski, Adaptive smart machining based on using constant cutting force and a smart cutting tool, Proceedings of the IMechE, Part B: Journal of Engineering Manufacture, 227(2), 2013, 249–253.
Author Response
Dear Sir/Madam,
We would like to thank you very much for your time and all the substantive comments to the text. They are very helpul and helped us to make our paper better. Plase find enclosed the detailed explanations and information about the changes we made to improve our paper.
Yours sincerely,
Antoni Świć, Dariusz Wołos, Arkadiusz Gola, Grzegorz Kłosowski

Reviewer 2 Report
The machining method proposed in this paper is very valuable for low-rigidity shaft turning.
Some suggestion is as follows:
(1)The description of paper should be further concise.
(2)The details of the experiment and the various results should be described further.
Author Response
Dear Sir/Madam,
We would like to thank you very much for your time and all the substantive comments to the text. They are very helpul and helped us to make our paper better. Plase find enclosed the detailed explanations and information about the changes we made to improve our paper.
Yours faithfully,
Antoni Świć, Dariusz Wołos, Arkadiusz Gola, Grzegorz Kłosowski

Reviewer 3 Report
Some comments that may help the authors to improve this manuscript are
- I recommend that keywords not included in the title of the article be used to increase the visibility of the article.
- Some variable names appear with a different and blurred font (example line 140 Fx1). This could be corrected.
- Figures 7 and 10 should be enlarged and their resolution improved.
- Figures 14 and 15. Shouldn't they contain legends?
- You could indicate the details of figures 19, 21 and 23 in figures 18, 20 and 21
Author Response

(The authors gave the same response as above.)
